# Daptomycin Pharmacokinetics in Blood and Wound Fluid in Critical Ill Patients with Left Ventricle Assist Devices

**DOI:** 10.3390/antibiotics12050904

**Published:** 2023-05-13

**Authors:** Stefanie Calov, Frederik Munzel, Anka C. Roehr, Otto Frey, Lina Maria Serna Higuita, Petra Wied, Peter Rosenberger, Helene A. Haeberle, Kristian-Christos Ngamsri

**Affiliations:** 1Department of Anesthesiology and Intensive Care Medicine, University Hospital Tübingen, Eberhard Karls University of Tübingen, 72076 Tübingen, Germany; petra.wied@med.uni-tuebingen.de (P.W.); peter.rosenberger@med.uni-tuebingen.de (P.R.); helene.haeberle@med.uni-tuebingen.de (H.A.H.); 2Department of Anesthesiology and Intensive Care Medicine, BG Trauma Center, 72076 Tübingen, Germany; fmunzel@bgu-tuebingen.de; 3Department of Pharmacy, General Hospital of Heidenheim, 89522 Heidenheim, Germany; anka.roehr@kliniken-heidenheim.de (A.C.R.); otto.frey@kliniken-heidenheim.de (O.F.); 4Department for Translational Bioinformatics and Medical Data Integration Center, Eberhard Karls University of Tübingen, 72076 Tübingen, Germany; lina.serna-higuita@med.uni-tuebingen.de

**Keywords:** daptomycin, cyclic lipopeptide antibiotic, left ventricle assist device, therapeutic drug monitoring

## Abstract

Daptomycin is a cyclic lipopeptide antibiotic with bactericidal effects against multidrug-resistant Gram-positive bacteria, including methicillin-resistant Staphylococcus aureus (MRSA) and vancomycin-resistant Enterococcus faecalis (VRE). For critically ill patients, especially in the presence of implants, daptomycin is an important therapeutic option. Left ventricle assist devices (LVADs) can be utilized for intensive care patients with end-stage heart failure as a bridge to transplant. We conducted a single-center prospective trial with critically ill adults with LVAD who received prophylactic anti-infective therapy with daptomycin. Our study aimed to evaluate the pharmacokinetics of daptomycin in the blood serum and wound fluids after LVAD implantation. Daptomycin concentration were assessed over three days using high-performance liquid chromatography (HPLC). We detected a high correlation between blood serum and wound fluid daptomycin concentration at 12 h (IC95%: 0.64 to 0.95; r = 0.86; *p* < 0.001) and 24 h (IC95%: −0.38 to 0.92; r = 0.76; *p* < 0.001) after antibiotic administration. Our pilot clinical study provides new insights into the pharmacokinetics of daptomycin from the blood into wound fluids of critically ill patients with LVADs.

## 1. Introduction

Perioperative antibiotic administration is important to avoid perioperative infections [1], but excessive use of antibiotics can be associated with anti-microbial resistance (AR) and toxicity (AT), which is a crucial problem for health worldwide and has been reviewed in detail by the World Health Organization (WHO) [2,3]. Antibiotic resistance caused by the inappropriate use of antibiotics is one of the top ten public health problems [4,5]. This problem is further exacerbated by the increasing presence of multi-resistant bacteria [6,7]. The presence of these multi-resistant bacteria and the inadequate use of antibiotics results in the compromised ability of antibiotics to control infections and, therefore, leads to unfavorable outcomes in critically ill patients, especially in patients with implants such as a left ventricle assist device (LVAD) [8].

LVADs have become an important treatment of choice for patients with end-stage heart failure, not only as a bridge to heart transplantation but also as a destination therapy [9,10]. Despite continuing improvements in surgery techniques, current LVAD therapies are not yet free from device-related infections [10,11,12]. However, due to the increased number of LVAD implantations, the number of complications has also risen, and therefore, the number of LVAD infections has also been elevated [12,13]. The guidelines for 2017 of the international society for heart and lung transplantation (ISHLT) recommend perioperative antimicrobial prophylaxis targeting Staphylococcus species, including methicillin-resistant Staphylococcus aureus (MRSA) [14]. Staphylococcus aureus and Streptococcus epidermidis colonize the skin, adhere to the driveline, and build a biofilm to survive, causing over 50% of all mechanical circulatory support (MCS) infections. Gram-positive bacterial infections are followed by Gram-negative bacterial infections, especially Pseudomonas aeruginosa (22 to 28%) and Enterobacteriaceae (2 to 4%) such as Klebsiella [14]. Previous clinical studies have observed the prevalence of infections and antibiotic treatment strategies after LVAD implantation but have not implemented therapeutic drug monitoring in this patient cohort [13,15]. However, a standardized perioperative antibiotic regimen for LVAD recipients is lacking.

Daptomycin is a cyclic lipopeptide agent and one of the last antibiotics to be developed to manage serious infections caused by Gram-positive multi-resistant germs [16]. Structurally and functionally, daptomycin can be related to the cationic anti-microbial peptides produced by immune cells, such as neutrophils, during acute inflammation [17]. Daptomycin induces a calcium-dependent membrane depolarization, which results in the loss of intracellular components, such as K^+^, Mg^2+^, and adenosine triphosphate (ATP). Infections with MRSA and vancomycin-resistant Enterococcus faecalis (VRE) are among the main indications [18,19]. Additionally, the US Food and Drug Administration (FDA) has approved the use of daptomycin to treat MRSA-associated bacteremia and endocarditis [20]. In recent years, due to its particularly high anti-biofilm activity, daptomycin has emerged as a treatment option for critically ill patients with implants, especially LVAD [21,22].

Therapeutic drug monitoring (TDM) is an adequate method to evaluate the blood serum concentration of daptomycin and can, therefore, be used to individualize anti-infective treatments and provide personalized medical therapy in intensive care medicine [2,23]. The primary goal of TDM is to reduce side effects due to toxicity in the case of overdose, especially when using drugs with a narrow therapeutic index (NTI) [24]. Critically ill patients with device infection, especially with device-related sepsis, undergo various pathophysiological changes, including alterations in drug distribution, metabolism, and clearance as well unexpected drug–drug interactions [8,25]. Facing these patients, it is not easy to calculate adequate antibiotic doses, and this leads to changing antibiotic blood concentration and results in sub- or suboptimal antibiotic concentration in the wound fluid or tissue of interest. Galar et al. revealed in a prospective study that a low daptomycin Cmin (<3.18 mg/L) is associated with poorer outcomes in patients with bacteraemia, complicated skin or soft-tissue infections, and endovascular infections [26]. A current metanalysis suggested that an elevated area under the curve (AUC) over 666 µg*h/mL is necessary for sufficient anti-the microbial effects of daptomycin [27]. Both parameters were evaluated in our pilot study to record under and overdosages in our LVAD recipients. However, a standardized daptomycin dosage regimen still remains missing.

The primary goal of our study was to determine daptomycin concentration in the drainage fluids and blood of critically ill patients with LVAD. Additionally, we compared the concentration of daptomycin in these two compartments to determine the penetration of daptomycin from the blood into the wound cavities. Further, we determined the correlation of daptomycin concentration in the blood serum and wound fluids one hour, 12 h, and 24 h after antibiotic administration. The secondary aim of our study was to record the infections that occurred to determine the drug’s efficacy after LVAD implantation in these patients. Additionally, we evaluated various clinical and laboratory parameters to detect the possible side effects of daptomycin treatment.

## 2. Results

### 2.1. Cohort Baseline Epidemiological Data and Clinical Characteristics

Between April 2017 and March 2019, 13 patients (three female and ten male) with an implanted LVAD were screened, and nine (one female and eight male) were included in our study (Eudra-CT-Nr: 2015-000125-36). Daptomycin was administrated prophylactically in all nine patients. Four patients were excluded from the study for various reasons (Figure 1).

All patients were, on the day of surgery, free from infection and, after LVAD implantation, received standard intensive care. Daptomycin was used prophylactically to prevent an early Gram-positive bacteria-related infection in patients after LVAD implantation. According to the internal standard operating procedure for LVAD recipients, therapy was discontinued after five to seven days if there was no evidence of infection. Blood and wound secretions were drained from the mediastinum and sometimes one or both pleural spaces in the first three postoperative days. All of the study participants’ baseline and clinical characteristics are presented in Table 1.

The median age was 61.1 (54–68) years, and most of the included patients were male. Concerning clinical presentation, we recorded an elevated acute physiology and chronic health evaluation (APACHE) and simplified the acute physiology score (SAPS). Furthermore, we evaluated the sequential organ failure assessment (SOFA) score. and as expected, we also recorded elevated SOFA values (mean ± SD: 11.2 ± 0.8) (Appendix A). All three clinical scores were elevated after LVAD implantation as expected in this critically ill patient cohort. A large retrospective trial postulated that the mean SOFA score was the best predictor of survival for LVAD recipients and may help to identify patients at an elevated risk of mortality in the early postoperative phase [28]. The same study emphasized that an elevated SOFA Score >12 was also associated with an increased 30-day, 90-day, and one-year mortality [28].

All LVAD recipients required short postoperative vasopressors and positive inotropic agents, such as norepinephrine and milrinone, during the observation time. Only one patient received the inotropic medication, levosimedan. Overall, there were no infections with evidence of Gram-positive pathogens, especially no infections of the catheter material or wound infections or around the driveline entry point. Two infections occurred in the nine patients included in the trial. A herpes simplex virus infection was detected by polymerase chain reaction (PCR) in the tracheal secretion of one patient; one episode of sepsis with evidence for Klebsiella oxytoca in the cultures’ tracheal secretion was detected.

### 2.2. Pharmacokinetic of Daptomycin

We evaluated daptomycin concentration in the blood samples and samples from drainage fluids at the indicated time points (Appendix A). As expected, over the three postoperative days, we observed increased daptomycin concentrations in the blood serum one hour after antibiotic administration, while antibiotic concentration in the drainage fluids were low one hour after intravenous administration. At 12 and 24 h after antibiotic administration, daptomycin concentrations were lower but still higher than daptomycin concentrations in the wound fluids. Daptomycin concentration in the drainage fluids were higher at the same time points on all three days than on the first day after antibiotic administration. We did not observe any accumulation of daptomycin in the wound drains on all three days (Figure 2).

The AUC of the concentration in the wound fluid was 19–54% (mean ± SD: 40% ± 11%) of the blood values; after 48 h, 23–64% (Mean ± SD: 40% ± 12%), and after 72 h, 32–63% (mean ± SD: 50% ± 10%). Further, we observed an adequate distribution of daptomycin from the blood into the wound fluids over three days. In almost all nine patients, we detected a high permeation of the antibiotic from the blood into the fluid wounds over the observation time. Only one case (Patient 2) experienced reduced antibiotic distribution over the three days. In one patient (Patient 4), daptomycin penetration was nearly unchanged over the follow-up period (Table 2). Next, we calculated the minimal (Cmin) and maximal concentration (Cmax) of daptomycin over three postoperative days in the blood serum. We observed raised daptomycin concentration in the blood serum after antibiotic administration, as expected, and a sufficient low daptomycin concentration over 20 (µg/mL) to avoid therapeutical failure [27]. The terminal half-time (t_1/2_) of daptomycin was documented previously for 7 to 9 h [29]. Interestingly, we observed an extended t_1/2_ up to 20.5 h over the 3 postoperative days (Table 2).

Next, we sought to determine the correlation between the daptomycin concentration in the blood serum and wound fluids. In our study, the Pearson correlation could not be used confidently due to the low number of samples or subjects. Therefore, we used the repeated measures correlation (Rmcorr), which is well-suited for analyzing paired and repeated measure data correlation [15]. The Rmcorr allowed us, based on constantly repeating values over three days, to bring these values together and to analyze them safely. Previous studies have revealed the changes in the daptomycin concentration in several ill patients, especially during sepsis or in patients with renal failure [29]. To our best knowledge, we presented at first a time-point correlation between daptomycin in the blood and the distributed antibiotic in the wound fluids. A statistical correlation analysis one hour after intravenous daptomycin administration failed to provide a significant correlation (IC95%: −0.356 to 0.641; r: 0.191; df 15; *p* = 0.462) (Figure 3A). We repeated the statistical correlation between the blood and wound fluids with daptomycin concentrations 12 h after intravenous antibiotic administration. The relationship between the blood serum and wound fluid daptomycin concentration at 12 h was statistically significant and almost linear, indicating the sufficient penetration of the antibiotics into the area of interest in the wound cavity (r: 0.862; *p* < 0.001) (Figure 3B). Next, we analyzed the correlation between the antibiotic values in the blood serum and wound fluid 24 h after daptomycin administration. Here, we observed a reduced but still significant correlation between the blood serum and wound fluid daptomycin concentration (r: 0.759; *p* < 0.001) (Figure 3C).

### 2.3. Safety Results

Daptomycin presented a low rate of adverse events or need for treatment discontinuation. No adverse events (AEs) or serious events (SAEs) occurred. No other rarer events were reported, such as rhabdomyolysis (Table 1), red-man syndrome, thrombocytopenia, eosinophilic pneumonia, or fungal infection. All nine included patients received the investigational medicinal product for the intended period of 5–10 days.

## 3. Discussion

Long-term cardiac and circulatory support by a left ventricle assist device replaces cardiac function during advanced heart failure [30,31]. Infections of LVADs complicate this treatment and adversely affect the outcome [8,32]. Gram-positive bacterial infections, mainly staphylococci and enterococci, are the most frequently isolated pathogens [33,34,35]. The continuous cutaneous defect by the driveline exit site represents an entry point for bacteria, increasing the frequency of LVAD infections [36,37]. Further risk factors, such as obesity (body mass index >30 kg/m^2^), diabetes mellitus, chronic kidney disease (glomerular filtration rate <60 mL/min/1.73 m^2^), history of previous mechanical circulatory support, and previous intensive care unit stay (>2 weeks), contribute to this vulnerability, especially for these patients [15].

Daptomycin, a bactericidal lipopeptide antibiotic, can be used to treat skin and soft-tissue Gram-positive infections [29]. The mechanism by which daptomycin kills bacteria is based on disrupting the bacteria membrane function and creating holes [38]. Daptomycin resistance and its possible mechanisms were described with clinical relevance but are still very rare [39,40,41]. Further, daptomycin serves as a salvage therapy when treatment with vancomycin has failed or is not permitted [42]. It is most useful in treating complicated skin infections, soft-tissue infections, or endocarditis with MRSA or VRE [18,24]. We used daptomycin regularly to prevent infections after LVAD implantation, as described previously [22].

Our pilot prospective clinical study examined daptomycin concentrations in serum and wound secretions after intravenous drug administration for three consecutive days. Daptomycin concentration in healthy subjects and population pharmacokinetic analyses have been widely performed and investigated in recent years [43,44,45]. ICU patients undergo various pathophysiological alterations, which affect the distribution, metabolism, and excretion of antibiotics. Furthermore, daptomycin dosing can be complicated by organ failure, sepsis, or drug–drug interactions [25,46]. To the best knowledge, we are the first to determine the daptomycin concentration in LVAD recipients. We observed a serum peak, reached after one hour, and gradually decreasing concentrations over the following 24 h. Our daptomycin serum concentration aligned with pharmacokinetic observations from previous clinical studies, including burn injuries and sepsis [29,47]. The typical serum pharmacokinetics of daptomycin have been observed already in previous studies and were confirmed by our serum findings [48,49].

Next, we evaluated daptomycin concentration in the chest cavities and determined the penetration ratio of daptomycin from the blood into the wound fluids nearest to the implanted device. To our knowledge, we are the first to determine the daptomycin concentration in wound fluids directly after LVAD implantation and to evaluate the daptomycin concentration this closely over three days. We detected adequate elevation in the daptomycin concentration of the wound area at 24 and 48 h, which were, on average, 40% in the blood, and after 72 h, this cumulated to almost 50%. In line with Wise et al., we observed a noticeably slower reduction in daptomycin concentration in the wounds [50]. On the contrary, Kim et al. examined an antibiotic peak after two hours and a rapid drop in daptomycin concentration in the wound fluids [51]. In contrast to Kim et al., we observed a noticeably slower reduction in daptomycin concentrations in the wound fluids. At present, there is no clinical study that has investigated the tissue penetration of daptomycin in patients with LVAD. We observed an adequate antibiotic penetration into the wound fluids nearest to the implanted device after the administration of multiple doses of daptomycin (6 mg/kg body weight). Only a few studies have evaluated the penetration behavior of daptomycin into soft tissues and bones during wound infections [51,52]. Both studies confirmed our findings that a 4–8 mg/kg body weight dose was sufficient to achieve an adequate concentration in the tissue.

The driveline exit represents a possible entering gate for various skin bacteria in a very vulnerable patient group [31,53]. Lambadaris et al. identified infections with Gram-positive bacteria as the main cause of an infection in the early phase after LVAD implantation [54]. Despite the fluctuating levels between blood and the secretion in the drains, at no time point did the minimum concentration fall below for the control of Gram-positive bacteria, such as enterococci and methicillin-sensitive *Staphylococcus aureus* [55,56]. Our study did not detect any skin or tissue infection with enterococci and methicillin-sensitive *Staphylococcus aureus*. Further, the daptomycin concentration we determined were also adequate to treat infections with MRSA. Fowler et al. and ESC guidelines for the management of infective endocarditis recommended a high dose of daptomycin (>10 mg/kg) to avoid possible resistance in patients [20]. However, Fowler et al. did not determine the concentration of daptomycin in blood or wound fluids and whether such a high dosage was necessary. Our pilot study highlights the importance of therapeutic drug monitoring in this critically ill patient population after LVAD implantation.

Various studies have demonstrated the efficiency of higher daptomycin dosages (>6 mg/kg) for the treatment of Gram-positive infections without an elevated risk of adverse effects such as increased creatinine kinase [57,58]. On the other hand, Bhavani et al. provided data that a Cmin <24.3 mg/L can be associated with a higher risk of toxicity and myopathy [59]. In our patient cohort, we observed elevated daptomycin Cmin values >24.3 mg/L on the second and third day of antibiotic administration and moderately elevated creatinine phosphokinase levels. However, we did not detect any other clinical signs that may support myopathy diagnosis. On the other hand, our LVAD recipients underwent operations on the heart, and during sternotomy, other muscles were also traumatized, which could lead to elevated creatinine kinase levels [60]. In this line, Bhavani et al. suggested that some creatinine kinase elevation may not be related to the daptomycin administration but may be associated with multiple causes [59].

### 3.1. Limitations

Our pilot clinical study had several limitations primarily related to the study design. The main limitations were our single-center population and the small participant size, limiting our findings’ generalizability. Another limitation of our study was the heterogeneity of the participants and the different locations, which were assessed to drain the fluid for our analyses. Further, an adequate control group was missing to provide possible differences between the observation and control groups.

### 3.2. Conclusions

In summary, our pilot trial demonstrated that daptomycin (5–8 mg/kg body weight) showed sufficient penetration into wound fluids drained from the chest cavities of critically ill patients with LVAD.

## 4. Materials and Methods

### 4.1. Study Design

This single-center prospective trial was carried out in the intensive care unit 39 of the department of anesthesia and intensive care medicine at the university hospital of Tübingen from April 2017 to March 2019. All patients scheduled for implantation of a left ventricular assist device were screened for inclusion and exclusion criteria. All patients received prophylactic antibiotic therapy, including daptomycin, during the first 5 or 7 postoperative days. Daptomycin (5 to 8 mg per kg of body weight) was infused over 30 min, once daily, at 7 a.m. Samples were taken following the schedule below (Table 3).

Blood samples were drawn from the arterial catheter already in place and considering hygiene requirements. The fluid samples from the drain systems were taken from the incorporated port under consideration of the standard hygienic procedures to avoid any contamination of the system. The samples were labeled with the number of the patient, the time of collection, and the date following the study protocol. All samples were centrifuged, and the supernatant was stored at −80 °C until dispatch to the laboratory in Heidenheim (Department of Pharmacy, General Hospital of Heidenheim). Measurements were performed using high-performance liquid chromatography with ultraviolet detection (HPLC-UV).

Further, we collected patient demographics, clinical and laboratory data, microbiological findings, the duration of anti-infective therapy, and clinical outcome information. The institutional review board approved the study, and written informed consent was obtained from all patients before participation.

### 4.2. Daptomycin Concentration in Blood Serum and Wound Fluid

Daptomycin (5–8 mg per kg body weight; MSD; Switzerland) was infused intravenously once daily as part of an anti-infective regime to prevent early infections of either the implanted pump or the driveline at 7 am. Blood samples were drawn from the arterial catheter already in place, and samples of wound fluids were taken from the drainage system directly from the incorporated port before they reached the container at indicated time points, considering standard hygienic requirements.

### 4.3. High-Performance Liquid Chromatography with UV Detection

The HPLC-UV measurement for daptomycin was conducted as part of the routine TDM program at the hospital of Heidenheim. Although data for certain target levels are sparse, there is an approach to reach AUC_24_/MHK >666 mg*h/L for effectivity and trough levels <24 mg/L and to minimize musculosceletal toxicity [61].

A detailed description of the HPLC method and validation process is included in Appendix A. Briefly, a gradient method carried out the separation of daptomycin in the serum and wound fluid on a C8 reversed-phase column. The method met the validation criteria stated by Valistat 2.0 (ARVECON GmbH, Walldorf, Germany) of the German Society of Toxicology and Forensic Chemistry (GTFCh). The method allowed for the easy and fast detection of daptomycin in human serum. Daptomycin peaks could be identified at a retention time of 9.9 min and at a wavelength of 370 nm. The calibration curve was linear over the concentration range from 5 to 100 mg/L, with a correlation coefficient of >0.99. The bias for accuracy within the samples analyzed on different days ranged between −1.6 and 1.6%. The lower limit of quantification (LOQ) was 2.5 mg/L. The limit of detection (LOD) was determined to be 1 mg/L. All acceptance criteria were applied and fulfilled for precision and for inter- and intraday assay performance. Inter- and intraday precision showed a low overall variation coefficient of 3.7 to 7.1%, indicating good assay performance. Mean recovery values were greater at around 100%. There was no evidence of interference for other drugs at the given retention time and wavelength.

A second substance (linezolid), the internal standard, was added to each sample to constantly check the measuring system. This way, whether the internal standard showed the same area in each chromatogram could be checked, providing proof of equal conditions in each measurement.

### 4.4. Statistical Analysis

All daptomycin values in the blood serum and wound fluids, from pleura and mediastinum, were examined among themselves and over time using a repeated measure correlation. Continuous measures were summarized as the means, standard deviation, medians, and interquartile ranges (IQR) according to the data distribution. The normality of the distribution was assessed by investigating skewness and kurtosis, as well as QQ graphs, box plots, and histograms. The association between blood and wound fluid daptomycin mean levels was examined using repeated measures correlation (R package rmcorr; version 0.5.4), which determined the relationship between two continuous variables while controlling for individual variances. The *rmcorr* correlation coefficient is a common statistical method for paired measurements assessed on two or more occasions for multiple individuals [62]. Similar to the Pearson correlation coefficient, the rmcorr coefficient (*r*_rm_) ranges from −1 to +1 and reports the strength of the linear association between two variables [63]. Linear mixed effect models assessed the difference in the mean daptomycin concentration in the blood serum during the three-time points with random intercepts; the slopes were set as fixed parameters. A two-tailed *p*-value below 0.05 was considered statistically significant. Statistical analyses were made using the rmcorr package in R statistical software version 3.6.4 and the SPSS version 27.0 (IBM SPSS Statistics for Windows, IBM Corp., Armonk, NY, USA).

## Figures and Tables

**Figure 1 antibiotics-12-00904-f001:**
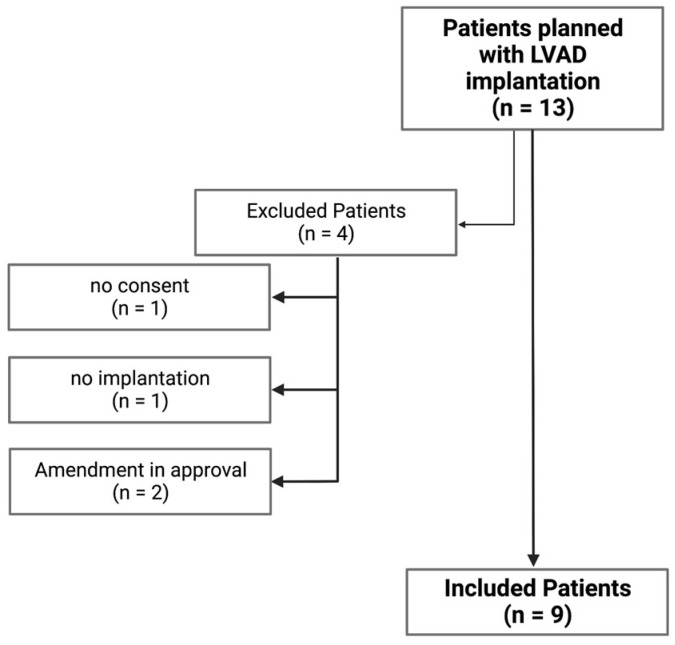
Schematic illustration of the screening, excluding, enrolling, and including of the study participants. Created with BioRender.com.

**Figure 2 antibiotics-12-00904-f002:**
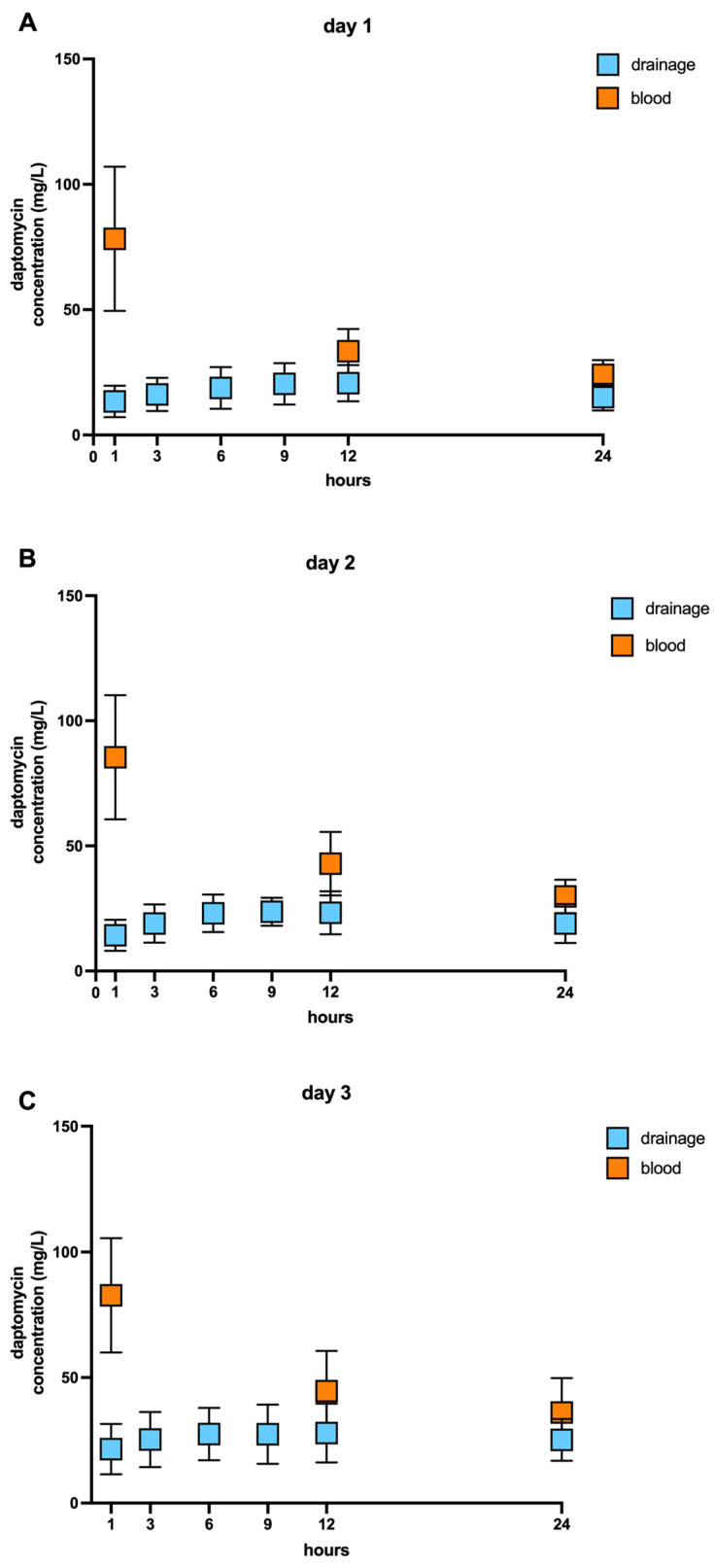
Boxplots of daptomycin blood serum and wound fluid concentrations (mg/L) over three days. (**A**) The blood daptomycin concentration (orange) was detected one hour after intravenous administration or (**B**) after 12 or (**C**) 24 h for three consecutive days. The blue boxplots represent daptomycin in wound fluids at the indicated time points (*n* = 9/each group). Data presented as mean ± SD.

**Figure 3 antibiotics-12-00904-f003:**
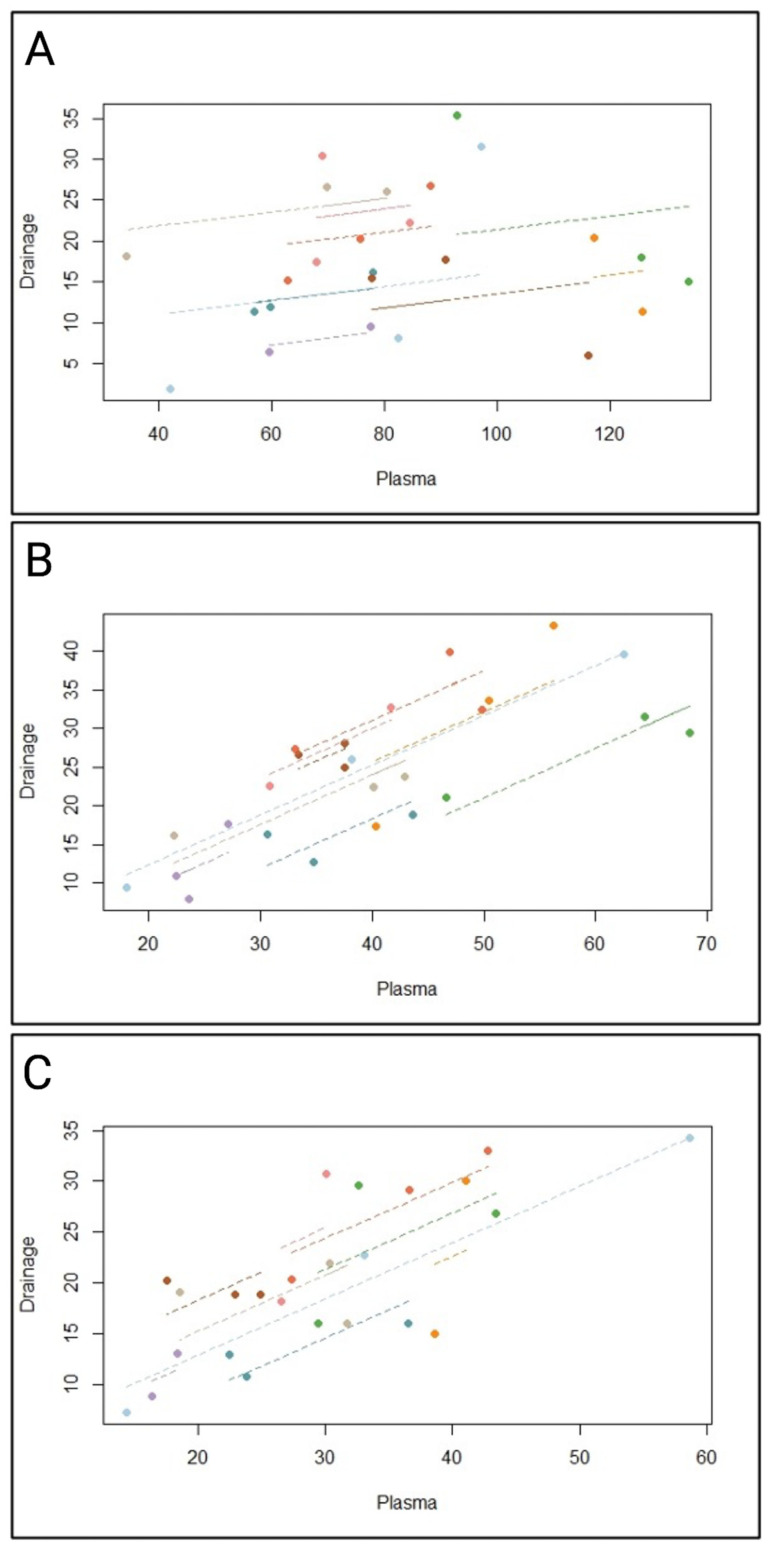
(**A**) Rmcorr correlation between the daptomycin concentration in blood serum (µg/mL) and wound fluid (µg/mL) after one hour, (**B**) 12, respectively, and (**C**) 24 h after intravenous daptomycin administration. Each point represents the correlation of each blood and wound fluid antibiotic concentration from all three postoperative days and each line represents the corresponding correlation.

**Table 1 antibiotics-12-00904-t001:** Baseline and clinical characteristics of the study participants. Data are presented as the mean and standard deviation (mean ± SD) or absolute numbers.

Characteristics of Participants	
Age (years/median/min–max)	61.1 (54.0–68.0)
Sex (female/male)	1/8
Daptomycin mg/kg body weight	
Dose adjustment d2	2 out of 9
Clinical Scores (mean ± SD)	
APACHEII Score	22.4 ± 1.3
SOFA Score	11.2 ± 0.8
SAPS Score	44.6 ± 4.4
Administration of vasoactive and inotropic medication	
epinephrine	3 out of 9
norepinephrine	9 out of 9
dobutamin	3 out of 9
vasopressin	4 out of 9
milrinon	9 out of 9
levosimendan	1 out of 9
Laboratory values	
creatinine kinase (U/L)	
postoperative	542.9 ± 170.7
day 1	631.8 ± 209.2
day 2	409.8 ± 193.8
day 3	362.6 ± 168.3
Myoglobin (U/L)	
day 1	407 ± 201
day 2	305 ± 256
day 3	410 ± 359
Lactatdehydrogenase (U/L)	
postoperative	366.3 ± 29.7
day 1	343.9 ± 23.1
day 2	299.8 ± 15.9
day 3	286. 8 ± 23.6

**Table 2 antibiotics-12-00904-t002:** Daptomycin pharmacokinetic parameter in blood and wound fluids 24, 48, and 72 h in patients after LVAD implantation.

	Day 1 (Mean ± SD)	Day 2 (Mean ± SD)	Day 3 (Mean ± SD)
AUC_blood_ _(µg*h/mL)_	1025 ± 334	1298 ± 478	1372 ± 464
AUC_drain_ _(µg*h/mL)_	383 ± 140	512 ± 173	695 ± 291
PR	0.4 ± 0.1	0.4 ± 0.1	0.5 ± 0.1
Cmin_blood_ _(µg/mL)_	24.1 ± 5.9	29.8 ± 6.7	29.9 ± 10.2
Cmax_blood_ _(µg/mL)_	78.3 ± 28.8	85.4 ± 24.8	82.8 ± 22.8
t^1^/_2blood_ _(hours)_	14.1 ± 4.0	15.8 ± 3.2	20.5 ± 6.7

AUC_blood_: area under the curve blood; AUC_drain_: area under the curve drainage; PR: penetration ratio (AUC_drain/_ AUC_blood_); Cmin_blood:_ minimal concentration blood; Cmax_blood_: maximal concentration blood; t_1/2blood_: terminal half-life_blood_.

**Table 3 antibiotics-12-00904-t003:** Schedule of samples taken from blood and drains.

Day 1						Day 2						Day 3					
Time to sampling (blood) after daptomycin infusion (h)																	
1		12		24		1		12		24		1		12		24	
Day 1						Day 2						Day 3					
Time to sampling (drains) after daptomycin infusion (h)																	
1	3	6	9	12	24	1	3	6	9	12	24	1	3	6	9	12	24

## Data Availability

The data presented in this study are available on request from the corresponding author.

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
