# Peer review of "Daptomycin Pharmacokinetics in Blood and Wound Fluid in Critical Ill Patients with Left Ventricle Assist Devices"

_antibiotics, 2023, doi:10.3390/antibiotics12050904_

Round 1
Reviewer 1 Report
This paper described Daptomycin pharmacokinetics in blood and wound fluid in Critical ill patients with left ventricle assist devices. Study is sound, I have some minor comments as under
Questions:
1. In abstract mention weather daptomycin was use prophylactically or as treatment?
2. Antibiotic use should be more defined? 36-44
3. Explain the type of infections that occur after implantation? 48
4. Explain the logic for taking samples at these time intervals? 80
5. What is the source of internal standard mentioned in line 98?
6. Were infected patients included in the trial? 118
7. Fig 5 a b and c need more clear representation?
8. Elaborate the method for take samples from the patient and also the safety concerns taken doing this? 212
9. Give reference in 309?
10. Give reason for using linezolid as standard reference? 318
Some minor changes required
Reviewer 2 Report
The discussion is well written and provides adequate context to the study issues, goals, and applications within the field. However, outside of this section, the manuscript is severely lacking in prose, content and organization. While this reviewer ultimately suggests a rejection, the question is pertinent, if not a start towards a hypothesis-generating idea. The authors should consult with a clinical pharmacologist and/or pharmacokineticist to bring the manuscript in line with useful clinical analyses and format/content.
General comments:
** There are several spots where a therapeutic level of daptomycin is said to be sought. Discuss this. Otherwise, you are discussing PK without a firm goal in mind. What level are you aiming? What/when should this sample be collected? To date, no guidance provides a TDM for daptomycin and the only strong data available is that subjects with low trough values (< 3) trended towards poorer outcomes. Cmax is known not to correlate with outcomes or predict other PK-based outcomes. If it is an institutional level, state this and provide further information on the daptomycin levels being sought.
** There is a mix of goals and aims stated throughout the manuscript, so the focus is hard to maintain. If you summarize everything, your cumulative goals seem to be (1) Correlation of plasma daptomycin to would fluid over time; (2) record infections and link to efficacy after LVAD implantation; and (3) side effects from daptomycin (clinical labs). 2 and 3 are not really discussed beyond the introduction and you have recruited so few subjects, you cannot perform a fair or meaningful analysis. Perhaps you could get more utility by making these questions retrospective within your particular institution to expand the available data available.
** I think you forgot to take out the guidance/instructional part of a few sections at the end.
** Supplemental file provided, but not referenced anywhere in the manuscript or description of its contents.
Specific Comments:
** Line 45: LVAD already defined previously.
** Line 49-50: Poorly worded. Please clarify this sentence.
** Line 62-64: Are LVAD implant infections common biofilm causing? Include this and I think there is a mix for line infections.
** Line 66-69: See comment above about an actual goal for TDM…not just an academic catch-all.
** Line 71-72: “… undergo various pathophysiological changes.” As this is a PK application paper, you should discuss these factors briefly and the implications on outcomes/therapy. Lots of refs here for daptomycin specifically too.
** Line 89-92: Just re-phrasing what is in Figure 1. Cut and consolidate.
** Line 97-98: “internal standard…” Is this the ‘internal standard of care’?
** Line 96-105: This paragraph has an odd organization going from describing daptomycin regimen, to question of daptomycin in fluids, then patient characteristics.
** Table 1: What about a summary of patient characteristics? Especially those important to daptomycin.
** Table 1: define ‘d1’, d2’, ‘d3’ in footnote.
** Line 111-113: Are these scores good or bad....give context to build your story. A good example is when people reference ECOG scores...we know 0-1 are subjects in relative good /uncomplicated condition.
** Lines 123-142: Very wordy and just describing the results in Figure 4 (and where is the figure reference????). Should focus on describing the results as trends and relevance (PK and/or clinical). Don’t just spew a bunch of numbers and they are shown in Figure 4.
** Line 150-157: Be scientific and use verbiage in the field. It is a comparison of daptomycin EXPOSURES within plasma and wounds to evaluate the PERMEATION (or DISTRIBUTION) of daptomycin into wounds. The authors need to have a pharmacokinetics-focused academic read and proof trier manuscript.
** Line 150-157: Hard to interpret. State the mean and give range in parentheses. SD or other items, like CV%, can be found in Table 2. And why is Table 2 not referenced???
** Table 2: Descriptive stats should be presented in the table...not just a table of raw data.
** Line 165-171: A correlation had better exist at 12 and 24 hours. We already know daptomycin Cmax (~1hr) doesn’t associate based on prior literature. Plus then at steady-state, so equilibrium and distribution is leveled out ... better give a correlation. Can you explain why you chose to sample at 1 hour? (based on what we know about daptomycin and the 12+ PK models of daptomycion in similar critically ill subjects).
** Line 171-172: A sentence fragment.
** Figure 5: Poorly labeled figure and legend does not note difference between A, B, and C.
** Line 192-194: You mention how these variables are important ...and yet not explored.
** Your Figure and Table number re-starts in Materials and Methods.
** Line 267: How were these times decided. How do they relate to the PK and/or disposition from what we already know about daptomycin?
** Line 272-273: Plasma or serum? Very different stability and bioanalysis issues known for daptomycin between these two matrices.
** Line 277: This figure does not provide any value to the manuscript. Remove it.
** Line 298: This figure does not provide any value to the manuscript. Remove it.
** Line 293-325: This is completely inadequate to describe the assay and its analytical robustness. There are zero details on the actual HPLC assay to know the chemistry. This is a very academic description, such as for a lay audience, but is insufficient to show that the levels were obtained in a systematic and robust manner. Overall bioanalysis is completely inadequately described. The authors need to reference a method paper for guidance on basic bioanalytical method reporting. You should find a bioanalysis/assay manuscript and use that as a template…or get a chemist to review your manuscript.
** Line 322-325: Just verification of linearity and precision are not 'validation'. Again, we cannot assume any analytical robustness has been evaluated as there is a complete lack of information on the assay.
There are several sentence fragments and verbage issues.
Reviewer 3 Report
Reviewer comments
In this study, the author conducted a well-designed study to investigate both the plasma and wound drainage pharmacokinetics (PK) of daptomycin in critical ill patients with left ventricle assist devices (LVADs). The manuscript is well-written and scientifically sound. The topic is relevant to the scope of the journal. The study outcome is clinically significant and potentially serve as the very first evidence to show the PK of daptomycin in wound drainage in this patient population. Additionally, the authors should be applauded for providing the raw data as a supplementary material for further investigation.
I have some comments for the authors to consider to further improve the quality of the manuscript.
Major:
1. The author should consider conducting a non-compartmental analysis (NCA) based on the data presented in figure 4. These PK data are valuable and difficult to collect, providing the basic PK parameters (Cmax, Cmin, t1/2, etc) derived using a NCA approach is useful to summarise the basic PK descriptive information of daptomycin in this patient population.
2. In the analysis shown in Figure 5, the author may consider testing a few covariates, such as the dose of daptomycin. This may be helpful to account for some of the apparent noises.
Minor:
1. Page 2 line 45, “Left ventricular assist devices” should be “LVAD” since the abbreviation has been defined in the last paragraph. Please check throughout the manuscript, LVAD should be used in all the following main context after the definition of this abbreviation.
2. Page 3, table 1, it seems age was described as “median (min-max)”, should have an annotation for that.
3. Page 3, table 1, looks like “2/9, 3/9, etc” in the table means 2 out of 9, 2 out of 9, etc. should have annotations for them for clarity.
4. Figure 4 legend, “The brown boxplots represent the 146 daptomycin in wound fluids at the indicated time points (n= 9/each group).” This should be blue instead of brown?
5. Figure 5 legend, the author should explain what the lines and colors represent. Additionally, please add corresponding (A) (B) and (C) in the legend following 1hour, 12 hours and 24 hours, respectively. Also, the authors should add units for both x and y axis on the plot.
6. Page 8, what is the association between “rhabdomyolysis” and Table 1? Please double check.
7. Pages 10 and 11 section 4.3, please indicate the UV wavelength for detecting daptomycin and linezolid, add appropriate references as needed.
8. Page 12, please add package version number for R package rmcorr.
Round 2
Reviewer 2 Report
I would like to thank the authors for their detailed responses to my previous comments. The manuscript is much improved by your efforts and now is interpretable by a far larger audience of clinicians and scientists. This is crucial given the early but important hypothesis-generating question you are describing.
While I have some qualms with the contents, this reviewer on introspection realizes these are more semantics and issues still being thought about by the clinical and academic communities at large in the TDM world. Thus, I have no further comments to provide on this manuscript.
Minor consistency and grammatical review would be appropriate (but common in copy edit phase of publication).